# The 3D Genome: From Structure to Function

**DOI:** 10.3390/ijms222111585

**Published:** 2021-10-27

**Authors:** Tapan Kumar Mohanta, Awdhesh Kumar Mishra, Ahmed Al-Harrasi

**Affiliations:** 1Natural and Medical Sciences Research Center, University of Nizwa, Nizwa 616, Oman; 2Department of Biotechnology, Yeungnam University, Gyeongsan 38541, Gyeongsangbuk-do, Korea; awdhesh@ynu.ac.kr or

**Keywords:** 3D, genome, topologically associated domain, cohesin, lamin, chromosome capture, hi-C, capture C, DNase, circular chromosome conformation capture, chromosome conformation capture carbon copy

## Abstract

The genome is the most functional part of a cell, and genomic contents are organized in a compact three-dimensional (3D) structure. The genome contains millions of nucleotide bases organized in its proper frame. Rapid development in genome sequencing and advanced microscopy techniques have enabled us to understand the 3D spatial organization of the genome. Chromosome capture methods using a ligation approach and the visualization tool of a 3D genome browser have facilitated detailed exploration of the genome. Topologically associated domains (TADs), lamin-associated domains, CCCTC-binding factor domains, cohesin, and chromatin structures are the prominent identified components that encode the 3D structure of the genome. Although TADs are the major contributors to 3D genome organization, they are absent in *Arabidopsis*. However, a few research groups have reported the presence of TAD-like structures in the plant kingdom.

## 1. Introduction

The genome, comprising both coding and non-coding DNA sequences, describes the genetic makeup of an organism [1,2]. The term ‘genome’ was coined by Hans Winkler in 1920, and it is now commonly used among researchers. Since the completion of high-quality reference genome sequences, we have witnessed several new discoveries in the ensuing decades, including genomic elements, structural and functional features of the genome, and genome organization [3,4,5,6,7,8]. An enormous amount of hierarchical compaction is required to produce three-dimensional (3D) chromatin structures from one-dimensional (1D) linear DNA sequences inside the nucleus under physical constraints [9,10,11]. The nucleus of a human cell contains 46 densely packed chromosomes [12]. In contrast, octoploid (*Opuntia*) [13], hexaploid (Sequoia) [14], and tetraploid (*Coffea*) [15] genomes contain 88, 66, and 44 chromosomes, respectively. However, *Ophioglossum* contains 1260 (decaploid, 630 pairs) chromosomes per cell [16], and these chromosomes can directly and accurately segregate themselves to the next cell during mitosis. Additionally, a ciliated protozoon, *Oxytrichatri fallax*, contains 1260–1600 chromosomes, commonly called nanochromosomes (amphidiploid) [17]. It is possible to organize the numerous chromosomes present in a cell into functional compartments at different genomic scales by folding them into hierarchical domains.

A chromosome has a distinct status in the nucleus, known as a ‘chromosome territory’, which is further partitioned into chromosomal compartments (A/B), topologically associated domains (TADs), and chromatin loops, mediated by the CCCTC-binding factor (CTCF; Figure 1) [18,19,20]. Chromatin folding plays a vital role in gene regulation, and transcriptional control is associated with physical contacts between target genes and the respective enhancers [18,21]. However, the functional loop between the genes and the regulator domain is predominantly carried out within TADs (Figure 2) [18]. High-level DNA folding and packaging generate extensive contacts between different genomic regions (Figure 2). These contacts can be in several forms, such as the folding architecture of proteins and chromatins and the proximity of DNA sequences to one another [22,23]. The packaging of chromosomes also brings them into contact with one another, as well as with the nuclear compartments, including the nucleolus and nuclear envelope (Figure 2) [24,25,26]. Cells progress through the cell cycle and undergo differentiation to form specialized cells [27]. The genetic information and function of a genome are not only associated with the epigenetic markers in the 1D linear DNA sequences, but also with their non-random spatial organization in the 3D nucleus. 3D chromatin organization is directly correlated with the functionality of the genome [28,29]. Chromosomes must undergo structural rearrangement leading to re-organization of contacts among the chromosomes (while maintaining the 3D structure of the genome) to influence transcription and function [30,31,32].

Further, nuclear mechanobiology is one of the most important mechanical properties for nucleus adaptability, which maintains a proper 3D-shaped nucleus to facilitate the organization of a 3D genome [33]. The 3D structure of the nucleus is determined by the interplay of the cytoskeleton–nucleus links, integration and composition of the nuclear lamina, and degree of DNA packaging in the nucleus (Figure 2) [34,35,36]. In the polarized shape of a cell, the apical actin stress fibers compress the nucleus into a flat ellipsoid shape [37,38]; however, in the isotropic shape, the relaxed, depolymerized actin results in the loss of mechanical tension, thus resulting in a spherical nucleus [39,40,41,42]. The contractile forces applied to the nucleus by actin filaments are counterbalanced by forces exerted by microtubules [43,44,45]. Microtubule and intermediate filaments reorganize to modulate nuclear morphology and deformability to provide a 3D-shaped nucleus [46,47,48]. The fine balance between the compressive forces exerted by actin and microtubules helps determine the nuclear morphology and regulate gene expression within the compact 3D genome [49,50]. However, the capacity for nuclear deformation is essentially based on the stiffness of the genome. Cellular chromatin exerts an outward entropic pressure on the nucleus, pushing it toward the nuclear envelope [51], which is counterbalanced by the post-translational modification of the histone tails, thus facilitating chromatin condensation [52,53]. Depending on the type of post-translational modification, genomic DNA is differentially organized into an open or condensed region (heterochromatin region) [33]. The level of chromatin condensation determines the size of the nucleus, coupled with the nuclear envelope and cytoskeleton [33]. Further, the extracellular matrix has been reported to control translational and rotational movements as well as fluctuations in the volume of the nucleus [54,55,56,57]. Changes in actomyosin contractility also cause alterations in nuclear morphology and modulate gene expression programs. However, the relative position of chromosomes in the nucleus depends on the 3D mechanical state of the nucleus [58]. Cells with a spherical nucleus preferentially align their chromosomes with the mechanical axis of the nucleus; however, in the altered shape of the nucleus, the nuclear axis remains perpendicular [33,58]. The chromosomes aligned parallel to the mechanical axis of the nucleus are more transcriptionally active than the others [58]. Therefore, mechanical regulation of the nucleus has the potential to change the chromosome structure and facilitate interaction between chromosomes and the nuclear envelope, and between genes and intermingling chromosomes.

Understanding how chromosomes are folded, packed, and positioned within the nucleus is of particular interest in deciphering the role of chromatin in gene regulation. Additionally, understanding the molecular distance between different genomic regions, or the molecular distance within the genomic regions and distinct nuclear compartments, can be of particular importance. To date, several methods have been developed to determine the architecture of a chromosome and its strengths and limitations. These include the chromosome conformation capture (3C) (Figure 3) [59,60,61] and high-throughput chromosome conformation capture (Hi-C) methods [29,62,63], which can be used to understand functional nuclear landmarks (splicing speckles and nuclear lamina) [18], chromosome territories [64], and TADs [65], thus facilitating the understanding of how frequently two genomic loci interact (Figure 2). DNA-FISH is a revolutionary method that allows visualization of chromosomes and genes in the nucleus [66,67]. This method provides single-cell information and allows only a small number of genomic loci to be analysed at a time. A 3C-based approach based on proximity ligation of DNA ends up being associated with chromatin contact (Table 1, Figure 3) [68]. However, the Hi-C map provides genome-wide chromatin contacts of kilobases to a few megabases [69,70,71]. The recent development of orthogonal ligation-free approaches, including genome architecture mapping (GAM) [72,73,74], split pool recognition of interaction by tag extension (SPRITE) [75,76], and chromatin-interaction analysis through droplet-based and barcode-linked sequencing (ChIA-Drop) [77,78], have revealed novel aspects of chromatin organization. SPRITE, GAM, and ChIA-Drop chromatin contacts identify topological domains and help predict complex chromatin contacts associated with three or more DNA fragments and uncover specific contacts that span tens of megabases [18,72,74].

## 2. Techniques to Study 3D Genome Organization

### 2.1. Microscopy-Based Visualization of the 3D Genome

The position and organization of chromosomes, domains, and specific loci in the nucleus have been studied using fluorescence in situ hybridization (FISH) [80,81,82]. However, FISH is limited to examining only a few predetermined loci [83,84,85]. It is a macromolecular recognition technology based on DNA or DNA/RNA’s complementary nature [86,87], where selected DNA strands are incorporated with fluorophore-coupled nucleotide probes to hybridize the complementary sequence [84,88]. For the hybridization, at least a single-strand probe must enter the nucleus [89,90] permeabilized using detergents or organic solvents (for example, methanol). The DNA is denatured by heat and formaldehyde treatment and is visualized through a fluorescence microscope to ensure the fine binding of a probe with its target [91,92]. DNA FISH is used to visualize chromatin compaction and the positioning of genomic regions within the nucleus [93]. It can map the physical distance between two or more differentially labelled genomic regions, thereby mapping the genes within the chromosome [18,66,94]. In addition, it can be applied to determine aneuploidy, microdeletions, microduplications, and sub-telomeric rearrangements [95,96,97,98,99]. The organization of the target DNA and nuclear compartments affects the accuracy and power of detecting different nuclear structures. We need to be sure that nuclear compartments are reserved during FISH. FISH probes are either synthesized oligos or generated through nick translation from a large DNA, resulting in overlapping fragments of 100–500 bp [100,101,102]. The probe may also cover the genomic sequence, from 30 kb to the entire chromosome [103]. The signal-to-noise ratio for locus detection increases with an increase in the length of the target because of the increased local fluorescence and high target specificity [18,104]. With standard 3D-FISH, long-range contacts within large genomic regions, including TADs or whole chromosomes, can be accurately detected [18,105]. However, short-range interactions of less than 100 kb are difficult to detect [18], making it difficult to quantify fine-scale chromatin folding below the level of TAD or the promoter–enhancer interaction. Cryo-FISH can provide high-resolution chromatin contacts, where FISH probes are hybridized to cryo-sections of cells [106]. Fluorescence is then visualized using a fluorescence or electron microscope [107,108]. The development of custom oligonucleotide arrays, including Oligopaints [109], can target 15-kb loci using conventional microscopy [102,109,110]. Oligopaints are libraries of synthesized oligonucleotides containing approximately 60–100 bp [111]. Subsequently, these oligonucleotides are amplified in a flexible manner using different primer pairs to generate FISH probes. Oligopaints can enable the study of chromatin folding in different epigenetic states at a resolution of tens of nanometers [112]. Oligopaint-based FISH, in combination with high-throughput imaging, can be useful for generating low-resolution contact maps, high-resolution contact maps (30 kb) for a stretch of 1.2–2.5 Mb, and maps for whole chromosomes [18,113].

### 2.2. Ligation-Based Detection of Contacts

3C is a one-on-one approach that can extract the chromatin interaction frequencies between two genomic loci via chromatin cross-linking and proximity ligation (Figure 3) [18,60]. Formaldehyde fixation is necessary for capturing protein-mediated and RNA-mediated contacts [60]. In the 3C method, the cells are cross-linked with formaldehyde, followed by fragmentation of chromatin using restriction digestion enzymes, such as HindIII or DpnII [114]. This is followed by proximity-based ligation of the adjacent DNA ends and determination of pairwise interactions using PCR sequencing. In the classical 3C method, a pair of interacting loci is interrogated using quantitative PCR, one at a time (Figure 3) [115]. This shows that 3C provides interactions between two loci and the required prior information of the target site (Figure 3).

Chromosome conformation capture-on-chip, commonly called 4C or circular chromosome conformation capture, is useful for interaction study of one region with the remaining part of the genome (one vs. all). The circular chromosome capture method, which is a part of the 3C technique, is used to address the existence of an epigenetically controlled network of chromosomal interactions. The 4C method is based on the principle of proximity ligation (Table 1), where the DNA–protein/protein–DNA generates a circular DNA molecule, using a high concentration of ligase and prolonged incubation for more than one week (Figure 3). Subsequently, reversal of the cross-linked primers proximal to the target sequence during ligation amplifies DNA with physical proximity, without prior knowledge of their identities. This procedure enables the amplification of sequences with a wide range of sizes in the cross-linked chromatin. 4C uses the same technology as 3C to obtain ligation products. The restriction product ligates with the 3C template and is incubated overnight, with frequent cutting of the second restriction enzyme (DpnII/NlaIII) [108]. Subsequently, linear sequences are generated to conduct primer hybridization as a 4C template. These 4C templates are hybridized to an array according to the standard immunoprecipitation (ChIP) protocol. The nuclear organization of active and inactive chromatin domains can be uncovered by the 4C principle [108]. Additionally, long-range cis-interaction of the SOX9 promoter can also be analyzed using 4C analysis [116]. Additionally, all contacts can be mapped at a single locus using the 4C principle. Primers for a region (promoter/enhancer) can be used to amplify all ligation partners of the locus, followed by sequencing of the amplified product (depth of 1–5 million reads per library) [117]. This helps to analyze the genome-wide interaction partner of the region of interest at a resolution of a few kilobases. This procedure is well suited for detecting short-range regulatory interactions, long genomic distances, and whole chromosomes [108,118].

Mapping of all the contacts within a large genomic region can also be performed by chromosome conformation capture carbon copy (5C) (Figure 3). In 5C (many-vs.-many), large genomic regions, up to several megabases, can be amplified from the 3C library using forward and reverse primers [119,120]. 5C has the potential to amplify large genomic regions at a sequencing depth of approximately 60 million reads per library to obtain a resolution of 15–20 kb for a 1 Mb region [121,122]. However, the resolution of 5C is highly dependent on the design of forward and reverse primers for all possible restriction fragments of a specific locus. If there is a lack of suitable primers, the mappable fragments must be excluded from the contact maps. The 5C technique overcomes the junctional problem associated with the intramolecular ligation step and is useful for studying complex interactions of specific loci of interest [123,124]. However, this method is unsuitable for the study of genome-wide interactions because it requires millions of 5C primers. Looping interactions can be studied using the 5C approach, and the approach can be widely applied for large-scale mapping of the cis–trans interaction network of the genomic element and a study of higher-order chromosome structures [59,119].

Super-resolution microscopy, such as stochastic optical reconstruction microscopy (STORM) [125] and photoactivation localization microscopy (PALM) [126], can enable direct visualization of the genome at the fine-scale level. A high-throughput imaging method (HIPMap) can be used to visualize endogenous genetic loci inside the 3D cell nucleus [127,128]. Furthermore, the HIPMap is useful for screening, validating 3C data, mapping translocation, probing DNA–protein interactions, and investigating the relationship between gene expression and localization [127]. Super-resolution microscopy is used to determine the structure of the chromatin fiber at the single-cell level, suggesting that nucleosomes are organized in groups and that nucleosome density is dynamic [129]. Although these revolutionary techniques are highly valuable for imaging, the current microscopy-based approaches are limited to a small number of genetic loci and do not allow a comprehensive analysis of the nuclear architecture of the complete genome. However, population-based 3C followed by Hi-C (high throughput derivative of 3C) (all-vs.-all) has the potential to study chromatin architecture (Figure 3). Hi-C uses DNA restriction fragments into which a biotinylated residue is incorporated, followed by blunt-end ligation, which allows ligation between the cross-linked DNA fragments [62]. This leads to the production of a genome-wide library of proximity ligation products in the nucleus. Each ligation product is tagged with biotin at the site of the junction, followed by shearing and subsequent pull-down of junctions with streptavidin beads [62]. The purified junctions are subsequently analyzed using a high-throughput sequencer to identify the interacting fragments [109]. This approach is suitable for detecting interactions of not only one viewpoint but also for detecting entire genomic regions or groups of targets [130]. Hi-C derivatives have been developed to further enhance the detection of chromosomal contacts. In Hi-C derivatives, the ends of cross-linked DNA restriction fragments are labelled with biotin and then ligated (Figure 3). Subsequently, the exonuclease activity of T4 DNA polymerase is used to remove the biotin label from the ends of the non-ligated fragments. The biotin-ligated fragments are then enriched with streptavidin beads to minimize the number of non-ligated DNA molecules in the sequencing library [63]. Approximately 50–70% of the sequencing reads are mapped to pairs of ligated restriction fragments in the Hi-C libraries. Using the tethered chromosome capture technique, a modified Hi-C derivative process, long-range intrachromosomal contacts are detectable [131]. Similarly, genome conformation capture (GCC) enables the sequencing of all DNA present in the 3C library [132]. The technique allows direct normalization of DNA abundance and removes any possible bias in sequencing or the presence of any possible genomic alterations. DNase Hi-C and in situ Hi-C are two other Hi-C derivatives (Table 1, Figure 3). In situ Hi-C omits the addition of sodium dodecyl sulphate, which is used during Hi-C. This omission in in situ Hi-C allows the ligation of chromatin fragments within the native environment of the nucleus, thereby reducing the number of random ligation events, leading to a reduced signal-to-noise ratio. This leads to a reduction in the sequencing depth, which enables the acquisition of high-resolution contact maps. The in situ Hi-C is faster, and convenient, as it does not require extensive dilution of the cross-linked chromatin prior to DNA ligation. Single-cell Hi-C enables us to detect contact maps in individual cells, thus allowing the study of rare cell types in the population [133]. This allows us to study the chromosome structure at specific stages of the cell cycle [134]. The single-cell Hi-C procedure uses in situ proximity ligation of cross-linked and digested chromatin, followed by the isolation of single-celled nuclei from the cell suspension and generation of sequencing libraries from each nucleus [64,133,134,135]. The single-cell combinatorial-indexed Hi-C method tags the DNA within the nucleus with a unique combination of barcodes [136]. The cells are fixed, lysed, and digested with restriction enzymes, followed by the separation of the intact nuclei onto 96-well plates, indexed with a specific barcode. Subsequently, the nuclei are pooled and separated again with a concomitant round of indexing. In situ proximity ligation and library preparation are performed on the pooled nuclei to generate single-cell Hi-C libraries. However, this method leads to inefficient recovery of contacts, inefficient digestion and ligation, and incomplete recovery of the input material.

Although 3C- and Hi-C-based techniques are remarkable for understanding chromosome folding in vivo, they have limited resolution due to noise and the coarse capture radius of restriction enzymes. Therefore, a Hi-C variant, known as Micro-C, has been developed to further resolve the structure [137]. Micro-C uses micrococcal nuclease to obtain mononucleosomes instead of restriction enzymes [138]. Micrococcal nuclease digestion is followed by mononucleosomal end repair and two steps of purification of the ligation products [120]. After purifying the ligation product, paired-ended deep sequencing is used to characterize the product. Micro-C enables the mapping of chromosome architecture at nucleosome resolution. It has the potential to capture chromosome folding, including compartment organization, interactions between CTCF sites, and TAD domains [137]. Additionally, it facilitates the understanding of a detailed map of the precise nucleosome position and localizes contact domain boundaries [119]. Micro-C exhibits a higher order of magnitude compared to Hi-C and allows the identification of approximately 20,000 additional loops in cells [137].

Another approach, known as proximity-assisted ChlP-seq (PLAC-seq), is also effective and useful for mapping long-range chromatin interactions [139]. It uses proximity ligation in nuclei prior to chromatin shearing and immunoprecipitation (Table 1) [139]. Fang et al. (2016) used PLAC-seq in mouse embryonic stem cells using antibodies against H3K27ac, H3K4me3, and RNA polymerase to resolve long-range chromatin interactions at the promoter and enhancer [139]. PLAC-seq has the potential to generate many long-range intrachromosomal and few inter-chromosomal pairs [139]. In addition, the technique has the potential to deduce chromatin loops with high sensitivity (8 times higher sensitivity than that of ChlA-PET) and specificity [139].

### 2.3. Non-Ligation-Based Detection

GAM is designed to analyze the 3D chromatin structure without the requirement of digestion and ligation [72]. It is based on the principle of linear genomic distance mapping to measure the 3D genome using ultrathin cryo-sectioning [140]. In GAM, cryo-sectioning of frozen fixed cells embedded in sucrose is performed in random orientation, followed by the generation of a single nuclear profile using laser microdissection. The nuclear profiles are subsequently subjected to sequencing, followed by sequence analysis. Once slices of the large collection of co-segregated possible pairs of loci in nuclear profiles are generated in random orientations, they are used to generate a proximity matrix of genomic regions. The GAM technique can map genome-wide chromatin contacts and is crucial for identifying topological domains. It can also detect highly complex chromatin contacts involving more than three DNA fragments and uncover specific contacts of approximately 10 Mb [18,72]. Further, GAM considers the spatial organization of chromatin architecture, including genome-wide contact frequencies, chromatin compaction, and the radial distribution of chromatin [72]. Beagrie et al. (2017) used 471 nuclear profiles of mouse embryonic stem cells using GAM procedures with a sequencing depth of 1.1 million reads per profile [72]. From this, they obtained 400,000 uniquely mapped reads per nuclear profile [72]. To understand the variation in detection, linkage disequilibrium was reported to be the best model to reduce bias. A comparison study with Hi-C and GAM revealed that they were highly correlated across the whole chromosome at a resolution of 1 Mb [72].

Sometimes, proximity-based ligation can fail to detect nuclear structures, such as interactions with nuclear bodies, as DNA regions may be far apart, making direct ligation difficult. This may create an incomplete understanding of genome organization. However, the development of SPRITE [76] has enabled genome-wide detection of higher-order interactions within the nucleus. In SPRITE, DNA, RNA, and proteins are cross-linked in cells, followed by nucleus isolation [76]. Subsequently, the chromatin is fragmented, and interacting molecules within the individual complex are barcoded using the split-pool strategy, followed by identification of the interactions via sequencing [76]. Once the sequencing is completed, all reads containing identical barcodes are matched [76]. In comparison to Hi-C, SPRITE can measure multiple DNA sequences that simultaneously interact with an individual nucleus to provide information about higher-order interactions [76]. SPRITE can also identify interactions across large genomic distances of >100 Mb [76]. This enables us to understand heterogeneous interactions far from the nucleus [76]. The SPRITE procedure does not require prior whole-genome amplification and hence is faster to perform. Because SPRITE does not use proximity ligation or whole-genome amplification, it can be directly incorporated into RNA molecules [76].

### 2.4. Cell Imaging of the Nuclear Structure

Chromosome folding is crucial for regulating proper gene expression and function, and it is a dynamic process that varies widely throughout the cell cycle [141,142]. TADs emerge as key players, leading to higher-order chromosome–chromosome folding, organization, and function through evolution. All these higher-order organizations are associated with tightly linked functional aspects, such as DNA replication and transcription. In relation to the genes associated with transcription, active genes are located more often toward the nuclear interior, even as the repressed genes are located toward the nuclear periphery (heterochromatic region) [143]. The chromosome-occupied distinct sub-nuclear territory is where the transcriptionally active loci are positioned at the surface. However, our ability to explore these genomic and chromatin dynamics has revolutionized technologies based on genome editing, which allows for simultaneous targeting of a particular locus in live cells. At present, genomic loci can be targeted in living cells using the clustered regularly interspaced short palindromic repeats (CRISPR) system, which uses an endonuclease-deficient form of Cas9 (commonly called dead-Cas9 or dCas9), fused with a fluorescent protein [144]. The tagged dCas9 is applied to the genomic loci through its interaction with sequence-specific guided RNAs. However, for concurrent labelling of the two genomic regions, the guided RNAs should be modified to function as a scaffold that brings the fluorescent protein to the target loci. However, the CRISPR system is well suited for repetitive genomic sequences because it relies on a single type of guided RNA.

## 3. Hierarchy of the 3D Genome

The folding of DNA into chromosomes has become a focal point in the study of the 3D genome [145,146]. The spatial positioning of genes for important biological functions, such as DNA replication, transcription, DNA repair, and chromosome translocation, is of particular interest. The folding of nucleosomes and chromatin remains a highly debated topic [147]. Although the folding of DNA into nucleosomes is well known, it is unclear how the two interact with one another. The folding of large and complex chromosomes requires a structural hierarchy of chromatin loops to genes, and enhancers to chromosomal domains and nuclear compartments [148,149]. Chromosomal territories are the most significant components where DNA becomes organized [150]. The chromosomal loci located on the same chromosome interact more frequently, even when separated by 200 Mb, than any two loci located on different chromosomes [151,152]. Chromatin loops facilitate interactions between the two chromosome loci in the same chromosome [153]. The nuclear envelope plays a key role in 3D genome organization, confining the genomic DNA into the 3D space [154,155,156]. The inner nuclear membrane is lined with a meshwork of lamin proteins, thus forming the NL [157,158]. The NL interacts with the lamin-associated domain (LAD) [159,160], and almost half of the genomes of cells are composed of LADs (0.1–10 Mb; 553 kb). However, not all LADs interact with the NL. In different cells, a few chromosomes are not localized toward the nuclear periphery, suggesting cell-to-cell heterogeneity. LADs are considered heterochromatic regions and are characterized by the presence of low gene density and lack of transcription [161,162]. During cell division and differentiation, some LADs lose their association with the NL [163,164], while others associate with the nuclear periphery. This leads to the altered gene expression, where activated genes move toward the nuclear periphery and inactivated genes move toward the interior [165,166,167,168], found in LADs. The NL serves as an anchoring location for the genome and constitutes a place for the heterochromatic loci that are scattered throughout the genome, connecting with it in three dimensions [169,170,171]. When NL associates with heterochromatin, nuclear pore complexes (NPCs) are enriched for association with euchromatin and active genes [172,173]. Thus, the nuclear envelope should be considered an organizing surface. Similar to the LADs, there are also 0.1–10 Mb (749 kb)-sized nucleolus-associated domains, which are co-localized to nucleoli or the NL [174,175,176]. Chromosomes of similar sizes and gene densities interact more frequently than those with dissimilar sizes and gene densities, and they interact in the nuclear space [177,178]. Short and gene-dense chromosomes group together near the center of the nucleus, even as long and less gene-encoding chromosomes locate near the nuclear periphery [167,179].

### 3.1. A/B Compartment

A study on Hi-C in combination with DNA fluorescence in situ hybridization (DNA-FISH) has led to a notion of chromosome territory and chromosome organization of the A/B compartment [180]. The A/B compartment is present on a scale of megabases [180]. The compartment can be estimated using DNase hypersensitivity sequencing, single-cell assay for transposase-accessible chromatin sequencing, and single-cell whole-genome bisulphite sequencing. Loci interactions usually occur between the loci within the same compartment. These compartments are sometimes cell-specific, although they do not comprehensively describe cell types in the genome [152]. Compartment A is usually associated with open chromatin, while compartment B is associated with closed chromatin. The A/B open or closed compartment can also be defined using chromatin immunoprecipitation sequencing for histone modification or DNase hypersensitivity assays [180]. Approximately 36% of the genome changes its compartment during the development of an organism. The compartments also change with changes in gene expression [180,181]. However, compartments A and B do not have a deterministic role in the specific pattern of gene expression. The estimation of A/B is usually performed via the eigenvector analysis of the genome contact matrix post normalization. The changes in the boundary between the two compartments occur where the first eigenvector changes sign [180]. The expected–observed analytical procedures normalize the bands of the contact matrix by deriving their mean to standardize the interactions between the two loci. The genome contact matrix normalizes to the first eigenvector, resulting in the A/B compartment. The identification of the compartment is reproducible and highly cell-specific [180]. The A/B compartment can also be identified using epigenetic DNA methylation data [180]. For this, the genomes are binned, and the average methylation values of CpGs and each bin are calculated. An elevated methylation level is reportedly associated with the open compartment (compartment A), whereas the closed compartment (compartment B) has little or no methylation.

### 3.2. Active Chromatin Hub

The development of several chromosome-capturing methods has been critical for the rapid progress in studying chromatin contacts, facilitating high-throughput mapping of contacts at different genomic distances at varying levels of resolution. Direct contact between multiple enhancers and promoters of the target gene leads to the formation of an active chromatin hub (ACH) by looping out of the intervening sequences that regulate the gene expression [182]. Mouse β-globin gene and its locus control region located 50 kb apart are brought into proximity in fetal-level cells but not in fetal brain cells [183,184]. The interaction sites of the β-globin locus cluster with multiple contacts simultaneously and through the formation of ACH [185]. The formation of ACH in the *β-globin* gene leads to its activation in erythroid progenitor cells [186]. The core subunit of the ACH includes the DNase I hypersensitive sites of the LCR, additional hypersensitive sites at the 5′- and 3′-ends, GATA1 transcription factor, cofactor LIM domain binding 1 (LDB1), and Kruppel-like factor [187]. LDB1 binds to the β-globin LCR and promoter, and promotes contacts between them.

The ACH hypothesis reveals how active genes in chromosomes are organized in the interphase nucleus. It has been well reported that enhancers are present at variable distances in highly expressed genes [188]. The histone modification and transcription factors associated with enhancers indicate that the enhancers are located at variable distances around highly expressed genes. The clustering of enhancers and the formation of ACH can greatly facilitate the folding of the chromatin region through the formation of loops. The formation of ACHs is most likely a function of LCR, which can be called the transcription-activating sequence [187]. Transcription-activating sequences are dominant over the chromatin position effect when inserted into different parts of the genome randomly, thereby creating a folding structure [187]. It has been reported that a single element from an LCR tandem repeat can lead to the creation of a dominant LCR transcriptional effect, suggesting that the contact driven by multiple transcription factors can alter chromatin organization by establishing an ACH loop [189,190]. A polymer model has been used to demonstrate the idea of chromatin looping through multivalent binders having an ‘on’ or ‘off’ switch behavior, which can enable the translation of a range of transcription factors into a single entity, with a chromatin-folding behavior. However, a link between promoters, enhancers, and silencers is yet to be established. In the case of the KIT locus, the promoter comes in contact with the distally located sequence and enhances transcription [191].

### 3.3. LADs

LADs are regions (0.1–0 Mb) of condensed chromatin that interact and bind with the NL at the nuclear periphery, thereby providing functional organization to the genome. The lamins are a mesh or a network of filament proteins called lamins A, B, and C [192] The lamin consists of an N-terminal head domain, a central coiled-coil rod domain, and a C-terminal globular domain. C-polymerization is associated with homodimerization, head-to-tail assembly of homodimers, and an antiparallel assembly of head-to-tail polymers into the filament [193]. The A- and B-type lamin undergoes maturation through C-terminal farnesylation, followed by tethering of the B-type lamin to an inner nuclear membrane, even as the A-type lamin is further processed, thus untethering it from the inner nuclear membrane. Lamin C does not undergo farnesylation, and lamins A and C are found in the NL and nucleoplasm, where they interact with the chromatin and regulate chromatin mobility [194,195,196]. Post-translational modifications of lamins, including O-GlycNAcylation and acetylation, contribute to the functional stability of the lamin network, thus reflecting the complexity of the NL and its role in the organization of higher-order genome architecture.

Genome organization is associated with the interaction of the active genomic region with the NPC, inner nuclear membrane, and NL through LADs. LADs maintain a higher order of genomic organization. They are rich in histone H3K27me3, and to a lesser extent, H3K9me2, which is part of the polycomb, repressed genes, and [197] heterochromatin. In the actively transcribed genes, the sequence of the open chromatin region loops into the interior of the nucleus. There are two types of LADs: constitutive LADs (cLADs, A-T rich) and facultative LADs (fLADs) [187,198]. During cellular differentiation, cLADs remain associated with the lamina, whereas fLADs are detached from the gene due to activation of the gene [192]. The inner nuclear membrane forms complex structures with lamins and helps to organize chromatin [199,200]. Post-mitosis, a few LADs relocate to the periphery of the nucleolus [201,202,203]. NADs located at the nuclear periphery also play important roles in this regard. Constitutive interaction of LADs with the lamina most probably acts as the structural backbone for the organization of interphase chromosomes. The mechanism underlying interactions among cLADs and fLADs is probably associated with DNA-binding factors [187,204,205], which are yet to be elucidated. Although LADs associate with low gene density regions, they contain thousands of genes, which are mostly not expressed, suggesting that peripheral nuclear localization is associated with gene silencing. However, Dam-methylated chromatin has revealed that the association of LADs with the nuclear periphery does not re-establish after a cell cycle [202]. A number of the LADs found in the nuclear periphery during one cell cycle can be found in the periphery of the nucleolus in the following cell cycle [202]. These dynamic LAD organizations in the lamina and nucleolar periphery suggest that both have the potential to organize a silent chromatin, although it remains unclear whether a similar gene-silencing mechanism operates in both compartments [187].

### 3.4. TADs

3D genome folding is highly organized, with a tightly linked process of DNA replication and transcription. The location of genes in chromosomes influences transcription; active genes are commonly located toward the interior, while repressed genes are localized toward the periphery of the nucleus [206]. All these aspects of gene and chromosomal organization require specific folding of chromatin contacts [207,208]. Genes are preferentially folded with intradomain chromatin interactions as clusters, compared to interdomain interactions [19,209]. These contact domains are commonly known as TADs [19]. TADs are architectural chromatin units that determine the regulatory landscape and shape the functional chromosomal organization [19,210,211]. They are characterized by long-range associations of the loci present between adjacent domains. This finding demonstrates that chromosomes are composed of strings of domains that are topologically separated from one another [149]. TAD sizes range from tens of kilobases to several megabases [149]. Genes located within the same TAD can bring coordinated dynamic gene expression during differentiation, suggesting the role of TADs in coordinating groups of neighboring genes. Although these TADs are well documented in metazoa, TADs and TAD-like structures have not been prominent in bacteria, fungi, and *Arabidopsis* [181]. However, TADs have been reported in *Oryza sativa* and cotton [212,213]. Bacteria, being prokaryotes, do not contain nuclei. Notably, TADs are conserved across the animal kingdom. They are separated by genetically defined boundaries, and the deletion of the boundary region in the X-chromosome inactivation center leads to a partial fusion of the flanking TADs [121].

### 3.5. CTCF and Cohesin

CTCF is a transcription factor protein associated with diverse functions, including positive and negative regulation of enhancers and X-chromosome inactivation (Figure 2). It binds to CCCTC nucleotides and is a single polypeptide of 727 amino acid residues [214]. It contains an N-terminal region, a central domain containing 11 zinc fingers, and a C-terminal region. CTCF binds to the nuclear matrix and stabilizes the nuclear architecture [215]. It also interacts with the nucleolus, and the interaction occurs via the phosphoprotein nucleoplasmin [216,217]. The role of this architectural protein is intriguing, as it mediates and blocks long-range interactions. However, the majority of CTCF sites are located within TADs, suggesting that CTCFs alone are not sufficient for boundary function (Figure 2) [121,218]. CTCF acts via its cofactor Yy1 for X-chromosome inactivation [214].

According to the loop extrusion model, extrusion of chromatin loops through the cohesion ring helps to organize the genome into spatial domains [219]. Cohesin is a ring-shaped protein complex that is topologically embraced by chromatid fibers. It mediates the cohesion between sister chromatids. Cohesin structural maintenance chromosome (SMC) complexes have been hypothesized to translocate along the chromatin, thus extruding the loop until it interacts with the CTCF and generates a chromatin domain to facilitate interaction between distant genomic regions [220]. Cohesin plays a significant role in gene expression and genome organization through the CTCF binding factor [219]. Although CTCF is a sequence-specific DNA-binding protein, cohesin associates with chromatids, independent of the DNA sequence signature motif (Figure 2). However, both cohesin and CTCF play mutual roles in bringing the sequence into proximity with chromosomes, as it is far apart from the chromosomes in the linear model [221,222,223]. Loop formation is an active process, and cohesin possesses ATPase activity and shares structural similarities with motor proteins, including kinesin and myosin [220]. Transcription leads to the movement of cohesin along the DNA to CTCF sites, suggesting the role of RNA polymerase in loop extrusion [224]. Additionally, CTCF works with the wings apart-like (Wap1) protein and regulates the distribution of cohesin over long distances, suggesting that CTCF and Wap1 play a regulatory role in genome organization.

### 3.6. Transcription Factors

Transcription and associated factors bind to DNA, resulting in transcriptional regulation [225]. Knockdown analysis of transcription factors GATA-1 and EKLF, or the GATA-1 interacting factor, FOG-1, has shown that the interaction between the globin promoter and enhancer becomes disrupted [183,226]. The association of locally recruited protein complexes can explain, in part or in full, long-range interactions of the distal loci. The interaction between transcription factors has the potential to affect chromosome conformation. Yin Yang 1 (YY1) is a ubiquitous transcription factor that binds the promoter and enhancer and mediates looping [227]. The cofactor protein, LIM domain-binding protein 1, is recruited to its target loci by transcription factors or cofactors. The structural maintenance protein encircles the DNA and pulls a loop of DNA through the ring [228]. YY1 and CTCF ubiquitously express transcription factors that bind to non-coding RNA [229,230]. The activities of these factors are also greatly affected by the organization of the genome. The DNA-binding activities of transcription factors are not only associated with binding to specific nucleotide sequences, but also with localization balancing within the nucleus [231]. The repressive heterochromatin region might inhibit transcription factor activity by physically hindering the factor and its machinery through the high density of the chromatin [232]. Most likely, transcription factors that enter the heterochromatic region become transiently trapped [233]. Nuclear architecture also provides temporal dynamics to transcription factor activity, as well as subsequent downstream transcriptional activities [227].

### 3.7. DNA Replication Timing in the 3D Genome

The genome of an organism is very complex, and organized in a 3D conformation by placing and regulating all the necessary genomic information in a specified manner. However, this complex genome undergoes duplication in each preceding cell cycle so that it can produce two identical copies of the same genome during the S-phase. Replication and duplication are ordered processes, and chromatins that experience early replication are rich in transcriptionally active genes in the euchromatic region; genes that are replicated late are enriched in the heterochromatin region [234,235,236,237]. These early and late replications are sometimes referred to as ‘open’ and ‘closed’, respectively, as per the 3C technique [238,239]. How DNA replication occurs temporarily and spatially across the genomes of the plant and animal kingdom remains unclarified. Lamins and geminins are absent in the plant kingdom, which are crucial for chromatin reorganization [240,241]. In addition, there are differences in the spatiotemporal distribution of replicating DNA between plant and metazoan nuclei; therefore, the DNA replication program in animals may not mirror that in plants [242,243]. A study of the replication timing program for the early, mid, and late S-phases in *Arabidopsis* chromosome 4 by Lee et al. (2010) reported no significant differences between these phases [244]. This finding led to the conclusion that the order of the origin of activation in *Arabidopsis* in the early and mid S-phases is stochastic, and the replication of euchromatin does not necessarily follow a strict temporal pattern. However, the same study group later observed differences between the early and mid S-phases in maize [245] and *Arabidopsis* genomes [246]. Approximately 41% of the genomes have been reported to show strong replication activity in more than one part of the S phase, reflecting the presence of heterogeneity in replication timing [246]. Some sequences replicate with high intensity in several portions of the S phase [246]. The distal part of the chromosome arm replicates earlier than the proximal part, whereas the centromeric and pericentromeric regions replicate last [246]. However, the short arms of chromosomes 2 and 4 replicate only in the mid– or mid–late phase, due to their proximity to the pericentric region [246]. Further, the early, early–mid, and mid segments are purely euchromatic, and that the mid–late and late phases are predominantly heterochromatic. This leads to the conclusion that replication timing may be independent of the local chromatin state (CS) or that the epigenetic states are not associated with CS analysis [246]. Wang et al. (2015) classified CS2 and CS4 as intermediaries between euchromatin and heterochromatin [247]. Because of the lack of transcription in CS2 and CS4, no enrichment of the histone mark associated with active transcription occurs. The study also revealed that read densities are significantly different in the early and mid S-phases, where early reads show that the high local maxima are separated by a deep trough, even as mid reads are distributed evenly with small dips and peaks [246]. This might be because early replicating regions have more origin and origin clusters than mid-replicating regions [246]. A comparative analysis between the *Arabidopsis* and maize genomes showed a similarity in replication timing signal, where the chromosome arms replicate early and the centromeric and pericentromeric arms replicate late [246]. However, more early-replicating regions exist in *Arabidopsis* than in maize, suggesting the differential presence of genic and nongenic (noncoding) regions (maize: 8% genic, 92% nongenic; *Arabidopsis* 51% genic, 49% nongenic) [246]. This finding suggests that genic sequences tend to replicate compared to nongenic regions. Furthermore, it has been reported that several dispersed blocks of mid–late and late replicating DNA exist in the maize chromosome arm, which are organized into the genic region separated by transposable element clusters [246].

## 4. 3D Genome and Gene Expression

Mechanical and biochemical signals perceived at the cell membrane activate transcription factors, which are subsequently directed to the target site to modulate cell- or tissue-specific gene expression [248,249,250]. When cells are placed on a substrate with different topographies, the nuclei change their shape, which leads to the activation of different gene expressions [251]. Cells placed on a different topography can exhibit distinct behaviors of proliferation, differentiation, and apoptosis [252,253]. Systemic turning of the contact area between the cell and extracellular matrix leads to altered gene expression of the matrix protein collagen [254]. Fibroblasts plated on polarized geometry express more matrix- or cytoskeleton-associated genes, whereas on isotropic surfaces, they express more cell–cell-junction and cell-cycle genes [251]. Nuclear architecture greatly shapes dynamic activities and expression of transcription factors, and with many genes, transcription occurs in bursts [255]. The transcription level is controlled by the burst frequency rather than the burst size [256]. Enhancer and promoter contacts, even in distant genomic loci, correlate with transcription, whereas the size of polymerase II correlates with the number of transcripts produced in a burst [257,258]. The temporal order of spatial clustering is a crucial aspect of gene co-regulation, and is necessary for activating various gene expression programs in different cell types [259,260,261]. During gene expression, genes are physically brought together. It may also involve the recruitment of transcription factors at different target sites and their subsequent clustering with other supporting transcription factor machinery [33]. It is highly possible that the integration and translation of biochemical cues into different gene expressions are enabled by different cellular and mechanical states [33]. The spatial organization of the genome has been optimized for cell-type-specific transcription, mediated by numerous mechanical and biochemical signals. Defects in mechano-signaling can lead to cell-to-cell contacts or an impaired extracellular matrix, which can lead to the disruption of the cytoskeleton–nucleus interaction, resulting in impaired nuclear morphology [262,263].

## 5. Data Structure of the 3D Genome

The close interaction between the 3D interphase DNA structure and gene expression has made chromatin folding a rapidly developing field of study. Several previous reports have been continuously challenged by the progress of research. For example, in the solenoid model, chromatin is folded into a 30-nm fibre, which is assembled into higher-order structures [264]. However, this report has not been substantiated when studied using electron microscopy tomography, which has shown highly distorted chromatin polymers [232]. We have discussed the role of TADs and the genomic contact and loop extrusion hypothesis based on CTCF and cohesin. It was previously thought that condensin compacts chromosomes by randomly bridging the DNA segment or supercoiling or passively pushing to sites of convergent transcription by RNA polymerase [220]. However, direct observation of loop extrusion in a single molecule has been found in condensin motor activity when quantum-dot-tagged yeast condensin was translocated along double-tethered DNA curtains [265]. The loop extrusion in the naked DNA was reported to be faster than that in DNA polymerase. While the loop extrusion of naked DNA translocation proceeded at a speed of 0.5–2 kb/s, those of DNA and RNA polymerases proceeded at approximately 1 kb/s and 1 kb/min, respectively [266,267]. The SMC complex uses ATP hydrolysis to perform the loop extrusion at a rate of 0.1–2 s^−1^ [220]. It has also been reported that cohesin- or condensin-binding factors possibly reduce the rate of chromatin loop extrusion. The major factors are the 11-nm nucleosome, RNA polymerase, protein complexes, and DNA structures [220]. However, the SMC complex can avoid these obstacles through nontopological binding, involving intermittent interactions with Nipbl1 and Pds5 proteins that alter the extrusion dynamics, where Nipbl1 possibly acts as a ‘dynamic safety belt’ for the cohesin protein [220]. A recent study has explored these aspects and reported the presence of TAD-like clusters even after cohesin knockout [268,269]. These results suggest that large cooperation of the architectural regulatory proteins, as well as the interplay of supercoiling, molecular binding, phase separation, crowding effect, and loop extrusion events, is needed. There are also questions regarding the functional units of chromatin and their hierarchy of folding, and the inner functional units of working TADs at the single-cell level. To understand all these intricate events, researchers have applied mathematical rules (stochastic, self-returning event) and studied a folding algorithm that can replicate experimental observations [270]. The most common type of chromatin interaction in the genome is that of the promoter and enhancer for transcriptional regulation and heterogeneous packing, which disperses local DNA accessibility and allows transcription and nuclear transport. From a polymer physics point of view, there is an apparent conflict between these two chromatin properties. It has been reported that chromatin resembles a fractal globule, which is a self-similar polymer in a collapsed state [271,272]. Although the fractal globule model observes high contact frequencies, it does not explain the spatial heterogeneity of chromatin packaging, which a 1D polymer cannot provide [270]. Therefore, researchers have provided a self-returning random walk (SRRW) mathematical model to address the contact-structure paradox [270]. It provides a non-branching topology of the 10-nm chromatin fiber and generates tree-like topological domains connected to an open chromatin backbone [270]. The SRRW model, presenting a new picture of genome organization, supports the hypothesis that local DNA density plays a critical role as a transcriptional regulator; the chromatin folds into a variety of minimally entangled hierarchical structures across the length from nanometers to micrometers without the necessity of a 30-nm fiber [270]. This model also explains the structure–function relationship of the interphase DNA with higher-order folding and a substantial reduction in dimension during genomic landscape exploration [270]. The model also predicts that the topological domains in single cells contain random-tree structures, where tree domains are regarded as nanoclusters and loops on a kilobase-to-megabase scale, serving as building blocks for large packaging domains. These tree domains are called ‘3D forests’ within the chromosome territory [270]. The size of a tree domain is positively correlated with the size of the genome, with considerable depression [270]. There is also a positive correlation between the tree domain size and packaging density, suggesting a size-dependent domain activity, where the nanodomain of the peak radius is approximately 70 nm [270]. Additionally, the model predicts a correlation between local DNA density and domain size, supporting the view that small domains are more active than large domains [270]. The first-order genome of a double helix DNA evolves to adopt a ‘virtual tree data structure’ for higher-order genome organization [270]. This tree-like topological domain is connected by an open functional backbone segment, which facilitates the proper organization of genomic contacts, package-based regulation of transcription, transport and accommodation of nuclear proteins, and transition between the interphase and mitosis [270].

A computational string and binder (SBS) model was proposed in polymer physics to understand the mechanism of chromosome compartmentalization, pattern formation, and chromatin folding [273]. According to the SBS model, chromatin folding can be driven thermodynamically by homotypic interactions between DNA sites that share compatible chromatin marks [273,274,275,276,277]. The chromatin filament acts as a self-avoiding walk string of beads, where specific beads function as binding sites for a cognate diffusing binder that can bridge them to allow folding [273]. The different binding sites can selectively interact with their cognate binder, and these binding activities can be computationally investigated by molecular dynamics simulations using Langevin dynamics with classical interaction potential [273,277]. This model explains chromatin folding thermodynamically by homotypic interactions between DNA sites sharing cognate chromatin [275,276,277,278,279]. This interaction takes place via protein binding to multiple sites, inducing phase separation of chromatin sub-compartments [280]. The association between chromatin sites and the nuclear lamina and speckles can also be inferred using the SBS model, with the help of the bridging protein transcription factor YY1, RNA polymerase II, and Polycomb repressive complex 1 [273,281].

The polymer model does not require any training or fitting against the experimental data and is sometimes used to test mechanistic theories of genome organization. This model focuses on global aspects of the genome, such as chromosome territories, rather than inferring specific interactions between certain loci [282]. In Hi-C, DNA fragments are captured as a library of DNA–DNA hybrids, followed by sequencing and alignment to a reference assembly. The Hi-C data are then analyzed by binning the count of DNA–DNA contacts from the aligned reads within equally sized bins across the genome [282]. This analysis generates a ‘contact matrix’ with C_ij_ (i = row and j = column) as the number of observed interactions between the DNA loci falling within the *i*th and *j*th bins [152,282]. The contact matrix depicts pairwise interaction frequencies between all pairs of DNA loci along the genome. The bin size is commonly referred to as the resolution of the experiment, as it provides the minimum scale to resolve the interactions [152,282]. The raw binned counts are normalized to correct biases, such as GC content, sequencing depth, and restriction fragment length. However, iterative correction and eigenvalue decomposition, or the ICE method, consider the total count of rows and columns of the contact matrix to be the same, thus solving the optimization problem. Optimization generates a ‘bias vector’ at individual genomic bin positions and is used to normalize the count matrix [152]. The contact matrices of raw or normalized interaction counts are plotted as ‘heat maps’ or contact maps [152]. The heat map data are either clipped or transformed to log, quantile, inverse hyperbolic sine, Pearson, or Spearman correlation [152]. Even with low-resolution binning and sparse data, Hi-C captures chromosome territories, thus generating a genome-wide intrachromosomal contact map [152]. In 3C-based techniques, the chromosome contact map is characterized by the enrichment of the interaction counts along the diagonal to capture short-range interactions in the linear genome. However, in 4C-seq, a vector of binned intrachromosomal counts corresponding to a single row or column in the Hi-C contact matrix exists. In the technique, the count decays exponentially as one moves away from the viewpoint [152]. Following the same principle, it is possible to plot average contacts versus genomic distance using Hi-C, which is commonly referred to as ‘contact decay profiles’, by which it is possible to understand the folding properties of the genome [152].

The Hi-C contact matrices are the major foundation for identifying 3D genome interactions, where the contact decay curve is used to interrogate the nuclear structure of the cell, as they can be generated using low-resolution data [282]. Accurate contact decay curves are important for determining the baseline probability of contacts at a given distance. Lieberman-Aiden et al. (2009) used Hi-C data to partition the genome into compartments A and B, where loci of one compartment interact more frequently among themselves than with other loci [152]. Subsequently, the compartments are divided into sub-compartments based on unsupervised Gaussian hidden Markov model clustering to high-resolution Hi-C data. Sub-compartmental analysis is routinely performed using low coverage and large bin sizes [282]. Deep sequencing and a high-resolution contact matrix are required to obtain fine contact features. Further, Dixon et al. used Hi-C data and a ‘directionality index’ to identify genome-wide TAD interactions, either upstream or downstream of genomic loci [218]. In all the 3C-based techniques (4C-seq, 5C, and Hi-C), the loci that make specific contacts are considered peaks in a plot of 4C-seq interaction counter vector, and peaks are identified as bins that show statistical significance with the theoretical model. The Fit-Hi-C method proposed by Ay and Noble (2015) uses genomic distance effects and ICE biases to model background resolutions [283]. An advanced model HiC-DC from Fit-Hi-C has been developed by modelling the sparsity based on the fact that genomic count data exhibit higher variance than expected [284]. Therefore, the HiC-DC method provides more statistically significant data [284]. In addition to HiCCUPS, the HiC-DC and Fit-Hi-C methods are also useful for studying loop contact [211]. HiCCUPS compares each entry in a contact matrix to different assemblages of surrounding entries to estimate the neighborhood using an (approximately) 5-kb contact matrix. A large number of CTCF loop contacts have been identified using the HiCCUPS method by Rao et al. (2014) [211]. The CHiCAGO method enables the analysis of Hi-C data. This method is based on the incorporation of the genomic distance effect and locus-specific noise model to identify interactions [285]. However, it has been reported that CHiCAGO, HiC-DC, and Fit-Hi-C use uniform genomic distance effects between equidistance loci [282].

## 6. 3D Genome Browser

The role of an enhancer that resides in the proximity of its target genes and the role of TADs are well known. The volume of the chromatin interaction data increases regularly, and efficient visualization and navigation of these data are the major bottlenecks for their interpretation. These factors make it a daunting task for an individual laboratory to store and explore them properly. To overcome these problems, several visualization tools have been developed, with unique features and limitations. The Hi-C data browser is reportedly the first web-based query tool to visualize Hi-C data as heat maps [152]. However, it does not support zoom functionalization and can hold only a limited number of datasets. The WashU epigenome browser visualizes Hi-C and ChIA-PET data, which also enables access to thousands of epigenome datasets from ENCODE and the Roadmap epigenome project (Figure 4) [286,287]. A Hi-C data matrix of files with large sizes up to hundreds of gigabytes tends to slow down the visualization process. Furthermore, it does not have the option to display inter-chromosomal interaction data as in a heat map. Hi-C data can also be explored using Juicebox [288] and Hi-Glass [289] at high speeds. However, none of these provides chromatin data, such as Capture Hi-C or ChIA-PET (Table 1, Figure 4). The Delta browser can display Hi-C data and a physical view of 3D genome modelling [290]. The 3D genome browser can help explore chromatin interaction data at the domain level and provide high-resolution promoter–enhancer interactions [291]. The 3D genome browser can facilitate zoom and traverse functions in real time, enabling queries using genomic loci, gene names, or SNPs (Figure 4) [291]. A user can also incorporate the UCSC genome browser with the WashU epigenome browser and query the chromatin interaction data with thousands of genetic, epigenetic, and phenotypic datasets, using the 3D genome browser [291]. Additionally, users can add or modify existing data or upload their genome or epigenome data, as well as view Hi-C data by converting the contract matrix into an indexed binary file called the ‘binary upper triangular matrix’ (BUTLR file). Users need to host a BUTLR file on an HTTP server and provide the URL to the 3D genome browser to obtain the full advantage of all the features of the 3D genome browser without the need to upload Hi-C data [291].

## 7. Conclusions

Understanding the 3D genome architecture can answer several fundamental biological questions regarding the integration of chromatin packaging patterns with various genomic and epigenomic features. We expect the development of enormous 3D epigenome and transcriptome datasets in the future. The identification of TAD boundaries facilitates the understanding of chromatin insulation. The roles of non-coding genetic contents (tRNA, rRNA, regulatory RNA, etc.) can be better understood using a 3D approach. The epigenetic 3D genome map enables the understanding and interpretation of the functional consequences of genetic changes associated with various diseases. Variations in structural variants can be detected using this approach.

## Figures and Tables

**Figure 1 ijms-22-11585-f001:**
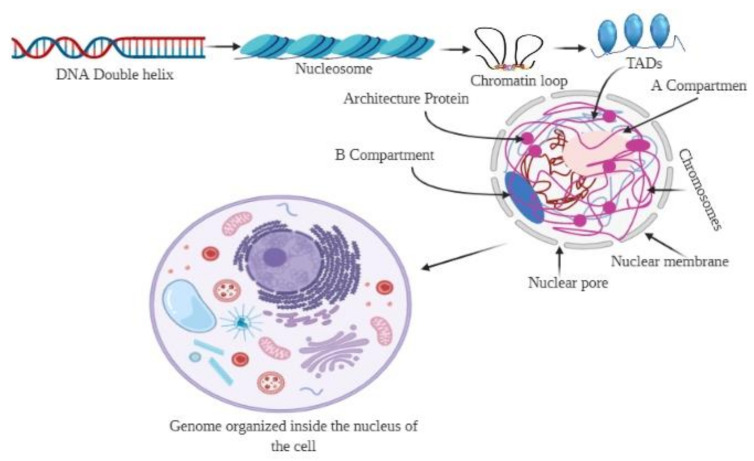
Organization of the 3D genome. The genomic DNA inside the nucleus possesses multiple levels of organizational structures. The primary structure, the linear DNA double helix, is packaged to form the secondary structural unit, nucleosome. The secondary structure brings approximately 7-fold compaction of genomic DNA. The 3D genome involves a higher-order organization in the 3D space of the nucleus, constituting topological features, including chromatin loops, A/B compartments, and chromosome territories. Chromatin loops are the basic building blocks for the 3D architecture of chromatins, while the topologically associated domains (TADs) are the basic structural and functional units of chromatins.

**Figure 2 ijms-22-11585-f002:**
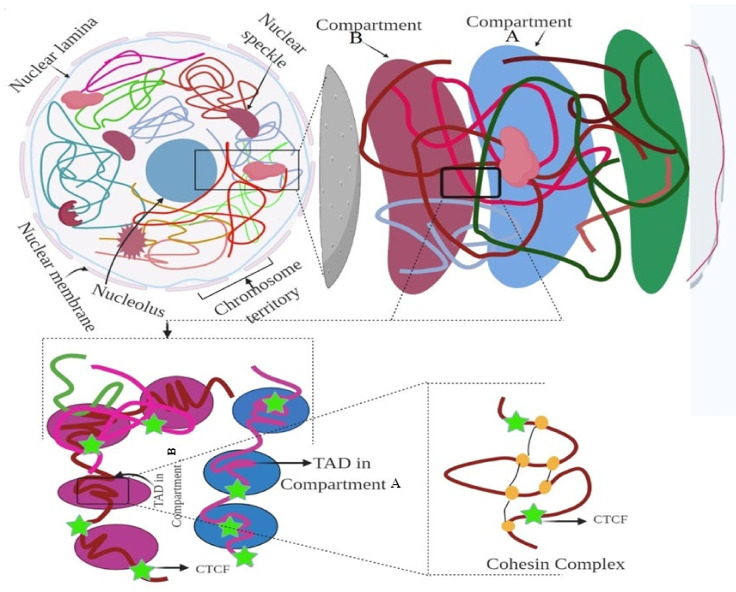
Hierarchical organization of the genome. Figure showing the nuclear compartment, CTCF, and TAD elements. The figure was prepared according to [9], with required modification.

**Figure 3 ijms-22-11585-f003:**
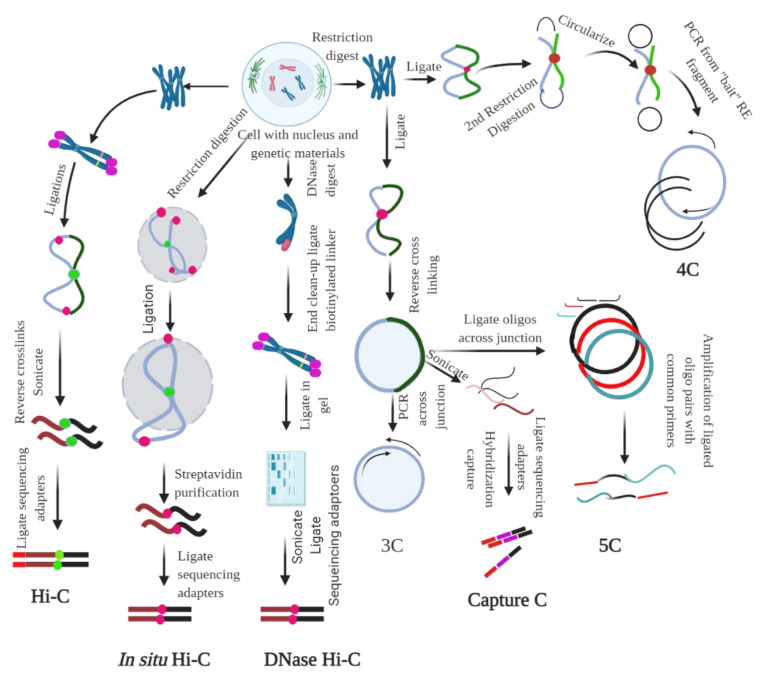
Chromosome conformation capture (3C) and its derivatives. It measures the contact frequencies of genomic loci by the proximity ligation of fragmented chromatin. All 3C procedures involve isolation of nuclei and DNA, followed by fixation of the chromatin. For Hi-C, ligation, followed by reverse cross-linking and addition of adaptors is required. For in situ Hi-C, streptavidin-based purification followed by the ligation of sequencing adaptors is required. For DNase Hi-C, the genetic materials are digested using DNase, followed by the ligation of biotinylated adaptor and adaptor-based sequencing. For 3C, restriction digestion is followed by ligation and reverse cross-linking and PCR across the junction. For capture C, reverse linking is followed by sonication, ligation of sequencing adaptors, and hybridization capture. In 5C, reverse cross-linking is followed by the ligation of oligos across the junction; the ligated oligo pairs are later amplified with a common primer. In 4C, restriction digestion and ligation are followed by the second step of restriction digestion, which circularize the genetic material; PCR is performed subsequently using the ‘bait’ RE fragment. The figure was prepared according to [18,79], with required modification.

**Figure 4 ijms-22-11585-f004:**
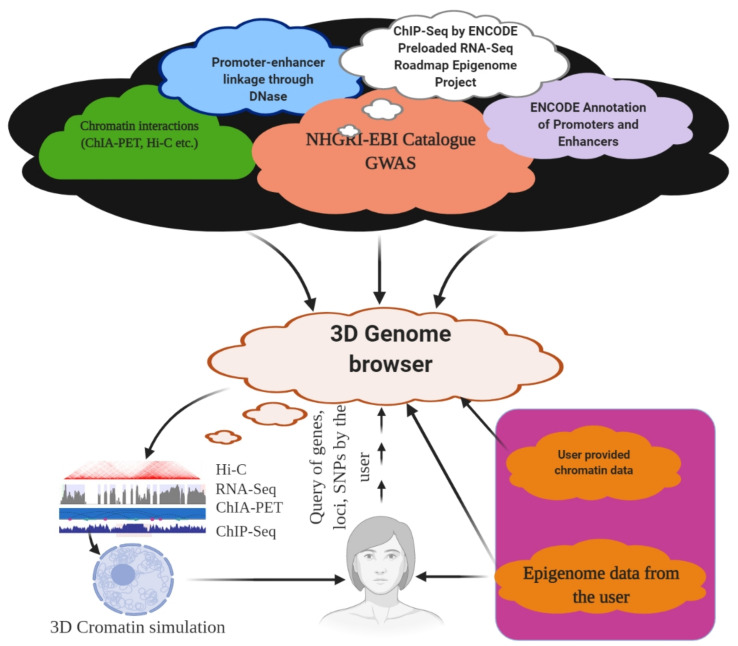
3D Genome browser. Using a 3D genome browser, it is possible to join multiple users worldwide to explore and understand chromatin interaction data, including ChIA-PET, PLAC-Seq, Hi-C, and capture Hi-C.

**Table 1 ijms-22-11585-t001:** Table representing different chromosome conformation capture method and its application.

Method	Assay Type	Ligation Procedure	Characteristics
snHi-C	Whole genome to whole genome	Proximity ligation	3C variant used to map chromatin interaction
scHi-C	Whole genome to whole genome	Proximity ligation	Hi-C variant enable to map chromatin interaction at single cell
sciHi-C	Whole genome to whole genome	Proximity ligation	Enable mapping of chromatin interactions using combinatorial barcoding
3C	One locus to one locus	Proximity ligation	Founding method of 3C
4C	One locus to the genome	Proximity ligation	Method to detect chromatin interaction between a specific locus and rest of the genome
Enhanced ChIP-4C	One to one gene	Proximity ligation	A variant of 4C. It improves the sensitivity through replacement of inverse PCR with primer extension
Unique molecular identifier-4C	Detect chromosomal interaction between loci and conditions	Proximity ligation	Improved 4C variant for improved sensitivity and specificity. It uses molecular identifier to derive high-complexity quantitative chromatin contact profiles
5C		Proximity ligation	Method used to probe chromatin interaction of multiple loci
CAPTURE	One to one in the region of interest	Proximity ligation & biotinylation	Uses biotinylated dCas9-mediated locus specific chromatin interaction
Capture-3C	Whole genome	Proximity ligation	High throughput 4C that combines with 3C with DNA capture technology
Capture Hi-C	Whole genome	Proximity ligation	High throughput 4C that combines with Hi-C with DNA capture technology
Dilution Hi-C	Whole genome to whole genome	Biotinylated proximity ligation	Maps topological domains whose boarders are occupied by CTCF binding sites
RNA-TRAP	Locus to locus	Proximity biotinylation	Combination of RNA-FISH with ChIP to probe chromatin interaction associated with transcriptional active genes
Targeted DNAse Hi-C	Whole genome to whole genome	Proximity ligation	Combines DNase Hi-C with DNA capture technology
Associated chromosome trap	Long range allele specific/interchromosomal	Proximity ligation	Used to identify distant DNA region that interact with defined DNA target
ChIA-PET	Whole genome to whole genome mediated by protein of interest	Proximity ligation	Combines ChIP with proximity ligation to detect genome-wide chromatin interaction mediated by specific proteins
PLAC-Seq	Whole genome	Proximity ligation	Proximity ligation conducted in nuclei prior to chromatin shearing
HiChIP	Whole genome/Multi-scale	Proximity ligation	Combines 3C with ChIP to ascertain genome-wide chromatin interaction intervene by specific protein
Hi-C	Whole genome to whole genome	Proximity ligation	Used to map all chromatin interaction in a cell population
DNase Hi-C	Whole genome to whole genome	Proximity ligation	Is variant of Hi-C that uses DNase I to break the chromatin
In Situ Hi-C	Whole genome to whole genome	Proximity ligation	Is an in-situ version of Hi-C that uses chromatin digestion and proximity ligation of intact nuclei
Tethered chromosome conformation capture	Whole genome to whole genome	Proximity ligation	Similar to Hi-C, but ligation performed in solid substrate rather than solution
In Situ DNase Hi-C	Whole genome to whole genome	Proximity ligation	Hi-C variant that uses DNase to break the chromatin
Micro-C	Whole genome to whole genome	Proximity ligation	Is a variant of Hi-C that uses micrococcal nuclease to digest the chromatin
Bridge linker Hi-C	Whole genome	Proximity ligation	Used to capture structural and regulatory chromatin interaction by restriction enzymes
Chromosome walks	Whole genome	Proximity ligation	Links multiple genomic loci together into the proximity
Genome architecture mapping (GAM)	Whole genome	Co-localization	Enables identification of the interactions of enhancer and active genes across large genomic distance
Split pool recognition of interaction by tag extension (SPRITE)	Whole genome/interchromosomal	Co-association	Enables understanding of genome-wide detection of higher-order interactions within the nucleus
Multi-ChIA	Locus to locus	Co-localization	Mapping of multiplex chromatin interactions with single molecule precision. Allow mapping of chromatin interaction mediated by protein of interest
Tethered conformation capture	Chromosome scale assembly	Proximity biotinylation	Allows mapping of inter and intrachromosomal contacts

## Data Availability

The article used all the previously published research and review articles present in the public domain and all can be available to the public.

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
