# Peer review of "The 3D Genome: From Structure to Function"

_ijms, 2021, doi:10.3390/ijms222111585_

Round 1

Reviewer 1 Report

After two rounds of extensive revisions, the manuscript has improved enormously, and I think it is now suitable for publication. In particular, the text clarity is now sufficient to convey information. 

I have only two remaining points:

1) The caption of Figure 2 is not matching the figure content. Precisely, the figure shows the hierarchical organization of the genome in territories, compartments, TADs, CTCF loops, while the caption is … the following:

“Figure 2. Gene expression in 3D genome. (a) Recruitment of transcriptional activator is sufficient to reposition a genetic locus located normally in the nuclear periphery towards the nuclear center. (b) on the right, tethering of transcriptional repressor to an active locus shift the TAD containing locus to the nuclear periphery. The absence of boundary elements due to genetic mutation or epigenetic mechanisms like DNA methylation can have consequences in gene expression. For example, (a) by bringing an active enhancer (pink) located in TAD (blue) close to the inactive gene leading to aberrant transcription of the gene. The transcriptionally active region is indicated in square mark in the interior of the nuclear space. Extended visualization of transcriptionally active region is depicted at the right (grey).”  

I already raised this point in my previous report, but it has not been addressed.  Please correct.

2) Rephrased sentence at line 474: “[…] between the two chromosome territories in the same chromosome.”

The authors probably meant to say: “[…] between two chromosome LOCI in the same chromosome.” In fact, each chromosome occupies a single chromosome territory by definition.

Author Response

Dear Reviewer

Thanks

Reviewer 2 Report

This is an attempt to review a very important and demanding topic in the current intersection of molecular, structural and developmental biology, namely the relationship between the eukaryotic genome structure and its function. This is not an easy task as there are many open questions, concepts that are difficult to present, a large number of experimental techniques involved and a very extensive corpus of  literature with often contradictory findings.

As this is no research work, that could claim to provide new findings, it doesn't fulfill its goal to provide a concise framework that would enable the better understanding of the 3D genome structure and function. This is already evident in the abstract which basically reads as a set of disconnected, random statements without focusing on a general question or concept. 

Many statements in the paper are over-simplified  (e.g. "the high level of DNA folding and packaging generates extensive contacts"), others are irrelevant (e.g. "The cell progresses through the cell cycle and undergoes differentiation, thereby giving rise to specialized cells.") and some are wrong (e.g.  chromosomes *need not* be re-arranged in order to facilitate transcription, changes in conformation in enhancer-promoter contacts are much more confined). Others, like "The role of no-coding genetic content (tRNA, rRNA, regulatory RNA etc) can be better understand using the 3D approach" (line 1039, p21) are grammatically and terminologically incorrect, while adding little to the discussion.

A large number of statements are false or make little sense, for instance "The chromosome enjoys a distinct status in the nucleus, commonly known as a “chromosome territory...". The territories are just one level of genome organization and far from constituting a general "status" that is acquired or lost at some point. 

Most of the figures (1,2 and 3) are adaptations of published figures from other papers, so it would be appropriate for the authors to cite the original works in the legends of the figures and discuss the degree to which the figure has been adapted.

Extended parts of the text contain no citations, even if they present concepts that are both novel and subject to heated debate. One clear example is Section 3. "Hierarchy of the genome" (p.11) which makes no reference to previous work.  Discussion on the concept of the Active Chromatin Hub (ACH) is also completely lacking references even though this has been the subject of an (uncited) review published 18 years ago (https://pubmed.ncbi.nlm.nih.gov/12971721/).

Most of the sections of the paper are not well-connected to each other and appear to contain fragmented information that is -in most of the cases- not well supported by references. They are, in addition, not well structured either. In discussing the hierarchy of the genome the authors go from chromatin hubs, to compartments, onto LADs and then down to TADs, which are much more confined and smaller-scale structural elements. The part referring to A/B compartments in particular is poorly written, lacking internal structure. The way A/B compartments are called is discussed in three different parts of the paragraph with no connection to each other.

The parts referring to CTCF, Cohesin, TFs and DNA replication are not related to the section "hierarchy of the genome" to which they ascribe. For the TF and DNA replication there are practically no connections drawn between their function and the genome structure. This strengthens the perception of a highly fragmented manuscript that appears to have been put together by stitching texts and as such does not serve its general goal. This is supported by the fact that such an extensive manuscript boils down to a 10-line-long paragraph titled "Conclusion", which contains little more than repetitive tautologies.

Less urgent but not insignificant points:
The paper needs to be re-read and edited. 
Parts of it are typed in coloured fonts (p 2, 10, 11, 12), quotation marks are left unclosed (p. 2, line 52). I am not sure if parts in red are supposed to represent highlighting of changes from a previous version of the paper.
Some proof-reading in the terminology is also required (e.g. there are no "chromatins" (plural)).

Author Response

Dear Reviewer

Many thanks for reviewing our article.

Please find the attached file to get the responses from the authors.

Regards

This manuscript is a resubmission of an earlier submission. The following is a list of the peer review reports and author responses from that submission.

Round 1

Reviewer 1 Report

In this review entitled “ The 3D Genome: From Structure to its Function” authors tried to described various aspect of genome organization varying from the techniques used to study the higher order chromatin organization, impact of 3D organization on genome regulation to the mechanism govern these structure in nucleus. However I feel the review article is written in many unorganized without following the story so it’s even hard to understand what they want to communicate, with lots or mistake which will mislead the reader and finally poor English and lots of grammatical mistakes throughout the article.

Here are some suggestions for the author to improve the article.

  1. In the article author mentioned six figures which they have not mentioned in the review article at a single place so obviously this can not be a mistake.
  2. Author fails measurably to cite the previously published works and correct citation. First whole paragraph of introduction has no citation.
  3. Line 51-53 not clear at all.
  4. Line 55-63 lots of reference information without any citation.
  5. Similarly line 66-76, statements without any citation and example. These are generalize hypothesis and need to be written in that way
  6. In introduction line 80-95, here author describe about the mechanical regulation of nuclear organization which is hard to understand.
  7. In line 101-012 author wrote 3C is based on DNA FISH. I am surprised how even one can write this which is completely wrong.
  8. In first paragraph of section 2.1, authors described the whole FISH process which is not relevant and even not written properly. Also they did not cite any reference for all the technical details mentioned.
  9. In line 169-171, ‘4C’ could not determine the interaction of one locus to whole genome, but it can determine the interaction of one locus with many (one to many).
  10. Line 172, 4C is not the part of the 3C but it is based on 3C.
  11. In line 222-223, authors wrote “The Hi-C uses biotinylated oligonucleotide complementary to a genomic region of interest that was used to pull down a specific ligation product” This is totally wrong. I am sorry but I am surprised that one can write the principle step of Hi-C wrong. In Hi-C, biotinylated nucleotide is used for end filling to label the interacting molecules.
  12. Line 252, tag the DNA with the nucleus ??. I believe it within
  13. Line 395-398, Hard to understand what author wants to say here
  14. In section 4.3, Even the definition of TAD is not correctly written (Line 401-402). Not true. TAD is a self-interacting genomic region.
  15. Line 407-410, author states that TAD or TAD like structure are not reported in plant however there are several study which showed TADs are not a prominent feature of A. thaliana genome organization but they have been reported for other plants including rice, maize, tomato and cotton
  16. Again in conclusion, author mentioned in the first line “absence of TADs in the plant genome” however TADs have been reported for plants including rice, maize, tomato and cotton.

Reviewer 2 Report

In this manuscript, Mohanta et al discussed the recent advances in the field of 3D genome, from technology development, computational tools to biological findings.

Main concerns:

  1. English language and styles require extensive editing.
  2. The organization of the text content is confusing. For one example, super-resolution microscopy was discussed in the Section “2.3. Cell imaging of the nuclear structure”, whereas there are two sections of visualization, “2.1. Microscopy-Based Visualization of the 3D Genome” and “2.3. Cell imaging of the nuclear structure”. For another example, the 3D structural features, active chromatin hub, Lamin-associated domains and Topologically associated domains were discussed together with chromatin structural proteins and transcriptional factors under the section of Molecular Players of 3D Genome Organization, but not in the section of Hierarchy of the 3D Genome.
  3. Although there are six figure legends, but there is no figure in the manuscript I got.

Reviewer 3 Report

This review about genome 3D organization covers many aspects of the field, from experimental technologies to map chromatin structure to the chromatin structural features discovered so far, the molecular players behind it, and the link with crucial functions such as transcription.

While I appreciate the attempt to put together such a comprehensive review, I think the authors have only partially accomplished the scope. In general, I have the impression that the authors have put together material with no clear guiding principle. Indeed, the review is unbalanced towards specific aspects while neglecting or completely ignoring others, and sometimes information is fragmented, confusing, or incomplete. Furthermore, the material is not well organized, the text and figures need to be largely refined, and the English language needs to be carefully checked.

In summary, I think the paper should be largely revised before it can be considered for publication in IJMS. In the following, I will try to clarify my points.

Main points:

1) Section 2 “Techniques to Study 3D Genome Organization” is unbalanced towards ligation-based technologies. It contains very detailed descriptions of 3C, Hi-C and their derivatives, while for example ligation-free technologies such as GAM or SPRITE are not even mentioned. Curiously, GAM and SPRITE are mentioned instead in the “3D Genome Browser” section as if they were visualization tools rather than technologies to map chromatin organization. Even among 3C technologies, some more recent advancements are not mentioned, such as the development of micro-C, which allows mapping chromatin at the nucleosome resolution (Krietenstein et al. Mol Cell. 2020, 78(3):554-565.e7, Hsieh et al. Mol Cell. 2020;78(3):539-553.e8). Technologies allowing to map chromatin contacts and specific proteins at the same time, such as PLAC-seq (Cell Res. 2016;26:1345–8.) and Chia-Pet (Nature. 2009;462:58–64), should also be mentioned. Finally, recent microscopy technologies, such as STORM, are curiously put in the “2.2. Ligation-based detection of contacts” subsection instead of in “2.1. Microscopy-Based Visualization of the 3D Genome”. And more recent advancements in microscopy should also be added, as those developed in Takei et al. Nat. 2021 and Su et al. Cell 2020. 

In summary, from a general review as the present one, I would expect a more balanced and comprehensive overview of experimental technologies.

2) Section 3: “Hierarchy of the 3D Genome”

Here I would expect a general description of the different levels of organization of the chromatin, from the formation of chromosome territories, A/B compartments, TADs, sub- and meta-TADs, loops, etc. Instead, the section in almost exclusively focused on the nuclear lamina and associated domains (LADs). Some of those arguments are present in the text but spread in other sections, and this is confusing. For example, TADs are described in the section “Molecular Players of 3D Genome Organization”, but TADs are structural features and not molecules. Also, there is here further subsection about LADs (4.2), which increases confusion. One solution could also be to merge sections 3 and 4 in one section and avoid redundancy as much as possible. Other fundamental features such as A/B compartment are not even explained in the text.

3) Section 7: “Data structure of the 3D Genome”

This section is about polymer models to understand the mechanisms behind chromatin organization, even though the title is misleading. Again, the section is biased towards one single polymer model (the SRRW model), among the many models proposed so far. The only other model mentioned is the loop extrusion, but at least a short description of the loop-extrusion process should be added here (LE is also mentioned but not explained in section 4.4), as well as a brief description of the chromatin features it can explain. Another main category of polymer models based on phase-separation mechanisms is not even mentioned. A brief description of those models and of the chromatin features they can explain should be added here (see e.g. Bianco et al. Curr Opin Cell Biol. 2020 Jun;64:10-17, for an overview about polymer physics models).  Finally, it would be worth mentioning the many data-driven, computational methods developed to infer information about chromatin 3D structure from experimental data (see e.g. Lin et al. Wiley Interdiscip Rev Syst Biol Med. 2019 Jan;11(1):e1435, about data-driven models).

4) Conclusion

It would be useful to add a proper conclusion section to wrap up the various arguments treated and indicate future perspectives. The current “Conclusion” is a section about the genome organization in plants… (which is not the main topic of this review, as far as I have understood).

5) Figures

Figures and figure captions are messy and should be largely rethought.

Figures 1,2 are ok, even though Fig.2 reflects the bias towards some 3C-based technologies.

Figure 3 is very confusing and I think it can be simply removed as I don’t see how it can be useful. In the first part of the caption, references should be added for the statements, which are not general and always true, about the relocation of a locus towards the nucleus center or periphery based on its activation state. The second part is very confusing (“The absence of boundary elements […]”), I don’t understand what the message is here and where I have to look in the figures to follow descriptions. About the figure itself, it is low quality and should be improved. I think there is confusion here between TADs and compartments. And I do not understand the meaning of items such as the 1 and 2 circled numbers, the blue circle, the semi-sphere below the blue region in panel b bottom, the zooms on the right (are they different from panel a?).

Figure 4. It is not clear what some items are, such as the grey semi-sphere on the right, and the green “compartment”. I would also improve the drawings of chromosomes, as they are messy. Some labels seem to be in the wrong position, as the chromosome territory. The bottom part is about TADs, and here I would not show compartments to avoid confusion (there is not a subdivision of TADs in A/B and it is not always true that a TAD is in entirely in one compartment). Also, the bottom right part about CTCF-cohesin is a confusing, not clear representation of the formation of loop domains.

Figure 5. This is again about TADs and could be removed or merged with Figure 4. But more importantly, this is a misleading representation of TADs, which are actually square-like by definition.

Figure 6. This is also confusing and should be better organized to give a clear overview of the available tools and data in 3D genome browsers. I don’t understand the criteria behind the figure, e.g. the categories of data in the various clouds, the directions of the arrows, etc. Just for example, why “epigenomic data from the user” go to the user? And why “chromatin data from the user” are written in smaller character than “epigenomic data”, and why they are separated at all. And what is the criteria behind the data categories in the top cloud? Some items are even useful like empty clouds… etc. etc.

Minor points:

6) Please add references to the figures throughout the paper

7) Please carefully check English and refine the writing

8) References are often missing in the text when citing a technology (just as an example, in section 2.2 there are no references for the many technologies reported, as 4C, TCC, DNase HiC, etc.) or a concept (just as an example, in Section 3 there is no reference for the various statements about LADs, which are not obvious). Please add references where needed. I see that the reference list is already long, but this can be optimized for example by citing a general review about the topic instead of many single papers (e.g., Ref.1 of the manuscript, Kempfer, et al., could be referenced for several technologies).

9) Line 41: “[…] the deletion of the boundary region in the X-chromosome inactivation center led to a partial fusion of flanking TADs”. This is only one example of the impact of genomic variation on the genome 3D structure. Indeed, this is a key and largely studied topic and a more general statement and reference would be suitable here, such as Spielmann et al. Nat Rev Genet 2018 Jul;19(7):453-467.

Round 2

Reviewer 2 Report

No figures again.

Reviewer 3 Report

The manuscript has been improved to a certain extent, but it still suffers from the issues I pointed out in my first review. 

Misleading information is still present. For example:

  • The representation of TADs in Figure 4 (old Figure 5 of the first version of the manuscript) is wrong. I already highlighted that in my first review, but the authors didn’t amend it. The paper they mention (https://www.futuremedicine.com/doi/10.2217/epi-2016-0111) correctly represents a TAD as an isosceles right triangle (indeed, it is half of a square, that is the way TADs are seen from Hi-C contact matrices). But in Figure 4 it is represented as a scalene triangle that is NOT correct. This makes me think that the authors have not even clear the concept of TAD.
  • Lines 97-98: this also doesn't seem right. 3C is not based on Hi-C but the contrary, and for sure 3C is not based on FISH.
  • Lines 104-105: What does “lenght” mean here? Maybe it is referred to Hi-C resolution? Precise technical terminology should be used.
  • Lines 227-228: How can a reduction in the sequencing depth allow to obtain higher-resolution contact maps? Maybe an increase?
  • Line 323: Chromatin loops are not known to facilitate interactions between different chromosomes.
  • Lines 415-420: Confusing sentences about TADs, further confirming that the concept is unclear for the authors themselves.
  • - Lines 632-633: chromosome territories are not seen from “intrachromosomal contact maps” but from inter-chromosomal contact maps.
  • Lines 667-668: it is not true that the role of a TAD is well known. On the contrary, this is still under extensive investigation.

Even if to a less extent than before, the manuscript is still not well organized. For example:

  • Compartments A/B are only described in the last section of the paper, “Data structure of the 3D Genome’, while they should naturally appear in the Section “Hierarchy of the 3D Genome”.
  • Details about the Hi-C output and its normalization also appear in the same section, while they could be better located, e.g., in the initial section ‘Techniques to Study 3D Genome Organization’. It is also weird that Hi-C contact matrices are only introduced at this point of the paper. Indeed, the authors extensively talk about TADs in the preceding sections, but TADs are detected precisely from Hi-C contact matrices.

English has been improved but the text is still not very well written, with several confusing sentences, e.g.:

  • Lines 57-59: “The chromosome has to undergo structural rearrangement […]_by maintaining the 3D structure of the genome” The end of the sentence is the negation of the beginning.
  • Lines 60-61: “This sentence would make sense if, in the previous sentences of the paragraph, they were talking about the importance of 1D DNA sequence and epigenetics. Instead, they were already talking about exactly the importance of 3D organization.
  • Lines 625-626: Normalization is not performed to ‘understand’ experimental biases but to correct them.
  • “Conclusion” section: the whole section is poorly written, even with grammar mistakes. An example for all: “The role of no-coding genetic content can be better understand using the 3D approach as the 3D position effect of non-coding genes are low.” The general meaning of the sentence is not clear. What is the 3D position effect? What does it mean that it is low for non-coding genes? And as far as I know “non-coding genes” don’t exist…maybe non-coding DNA?  And there are several grammar mistakes: “No-coding” --> non-coding, “Understand” --> understood, “are low” --> is low. Similar considerations can be done for the other sentences in the Section and throughout the paper.

Other specific comments:

- There must be a mistake with the caption of Figure 2 (about the genome 3D organization). Indeed, it is not referring to the content of the figure. I guess this is instead the old caption of a figure removed in the revised version.

- Some references should be added in the first part of Section ‘Hierarchy of the 3D Genome’ (lines 314 -342).

- Lines 297-298: Not clear what the term ‘evolution’ means here. Maybe temporal evolution?

To summarize, at least the points above should be addressed in order for the manuscript to be sufficient for publication. By the way, with the material they collected, the authors could have written a much better review than this one, and it is a pity that their effort is lost. I would suggest that the authors take more time to better digest all this material and then rewrite the paper from scratch with more awareness and resubmit it to this or another journal. As a further, more general suggestion to make the review more useful for a reader, I would either remove some unnecessary details (for example, in the description of experimental techniques) making the review more concise while keeping it comprehensive; or, alternatively, I would focus on a more specific aspect (for example, experimental techniques to map the 3D genome) and give full details.